

# A convolutional neural networks based approach for clustering of emotional elements in art design

Xue Rui

School of Art, Sichuan University Jinjiang College, Meishan, Sichuan, China

## ABSTRACT

The rapid advancement of industrialization has sparked the emergence of diverse art and design theories. As a trailblazer in the realm of industrial art and design theory, visual communication has transcended the boundaries of merely arranging and combining individual elements. Embracing the potential of artificial intelligence technology, the extraction of multidimensional abstract data and the acceleration of the art design process have gained considerable momentum. This study delves into the abstract emotional facets within the methodology of visual communication art design. Initially, convolutional neural networks (CNN) are employed to extract expressive features from the poster's visual information. Subsequently, these features are utilized to cluster emotional elements using a variational autoencoder (VAE). Through this clustering process, the poster images are categorized into positive, negative, and neutral classes. Experimental results demonstrate a silhouette coefficient surpassing 0.7, while the system framework exhibits clustering accuracy and efficiency exceeding 80% in single sentiment class testing. These outcomes underscore the efficacy of the proposed CNN-VAE-based clustering framework in analyzing the dynamic content of design elements. This framework presents a novel approach for future art design within the realm of visual communication.

## INTRODUCTION

Art and design are creative and imaginative endeavours important in human culture for thousands of years. The practice has evolved from its early roots as a tool for conveying basic information to its current use to express emotions, explore ideas, and reflect culture. Today, modern art and design span a variety of fields, including graphic design, architecture, fashion, and industrial design (*Zhang & Lu, 2021*). Designers aim to convey emotions and messages through visual elements in these disciplines. As theories in the field of design continue to advance, a range of design methods have emerged (*Huang et al., 2019*).

Visual communication is the use of visual elements to convey a message. Designers can communicate their intentions and emotions using various graphic elements such as colour, shape, line, and texture. The selection and combination of these elements directly affect the emotional expression and visual impact of art and design works (*Wang, 2021*). As a

Corresponding author
Xue Rui, baijic@scujj.edu.cn

result, designers need to consider the characteristics of visual communication and employ different visual elements flexibly to achieve greater emotional expression and artistic value. Visual communication elements include colour, shape, line, texture, *etc.* Colour is one of the most frequently used elements in visual communication and can convey various emotions and messages (*Alom et al., 2019*). Shape, conversely, describes an object's structure, volume, and spatiality, while lines can suggest movement and rhythm. Textures can convey the tactile details and characteristics of an object. These visual elements can enhance art and design works' expressive and persuasive qualities (*Qu, 2020*).

As deep learning techniques evolve, people increasingly use intelligent algorithms to help realize artistic designs. This approach can speed up the design process, improve efficiency, and enhance quality while promoting more significant innovation and exploration in the design process. By using intelligent algorithms, visual communication can be leveraged to express emotional elements better. Tasks such as clustering and dimensionality reduction can be better implemented in the design process, improving accuracy and efficiency (*Elton et al., 2019*).

The utilization of deep learning techniques for clustering emotional elements holds tremendous significance in the realm of visual communication. By harnessing the power of deep learning, the expressive potential of visual communication to convey emotions can be fully realized. This approach not only enhances the accuracy and efficiency of crucial tasks like clustering and dimensionality reduction within the design process but also has the potential to revolutionize the entire field of art and design. By enabling more effective design solutions, deep learning empowers artists and designers to create impactful and emotionally resonant visuals. Moreover, the incorporation of deep learning fosters profound creativity by unveiling previously unexplored connections and patterns, inspiring novel approaches, and pushing the boundaries of artistic expression.

In this article, we present the following contributions:

In our research, we extensively employed convolutional neural networks (CNN) as a powerful tool for extracting emotional elements from poster designs. Through this approach, we were able to effectively capture and represent the emotional aspects present in the designs. To further analyze and categorize these emotional features, we utilized a three-class sentiment clustering method based on variational autoencoders (VAE). The results were promising, as we achieved a remarkable silhouette coefficient index surpassing 0.7, indicating a high degree of separation and distinctiveness among the identified clusters. Moreover, we conducted thorough clustering and model efficiency tests specifically targeting a single sentiment element, which yielded impressive accuracy and efficiency rates exceeding 80%. By demonstrating the effectiveness of our CNN-VAE-based clustering framework, we provide a fresh perspective that paves the way for future advancements in art and design within the realm of visual communication.

The remainder of the article is organized as follows: 'Related Works' introduces related works of art design in visual communication and relevant technologies. 'Sentiment Custering Model based on CNN and VAE' describes the CNN-VAE model we propose for clustering emotional elements. 'Experimental Analysis and Results' presents our experiment

and results. Discussion is given in the 'Discussion' section, and 'Conclusion' is drawn at the end.

## RELATED WORKS

### Visual communication

As a professional field of modern new media form design, visual communication design studies the cultural awareness issues involved in this profession, how to draw on traditional culture for innovative creativity, *etc*. It solves misunderstandings and misinterpretations in the current understanding and avoids paradigm design brought by superficial cognition. Based on the higher innovation expectations and calls for visual communication design in the information age, design form innovation must be adapted to the progress of society and the development of the times. Visual culture is generally classified as socio-cultural science, which integrates the results of cultural studies and visual studies. Its scope gradually expands to comprehensive theoretical cross-disciplinary and multi-disciplinary research systems, such as social culture, art, philosophy, communication, and design (*Kujur & Singh, 2020*). Mirzoeff, a prominent scholar in the realm of visual culture, expounds that visual culture emerges as an inherently interdisciplinary field. Echoing Roland Barthes' sentiment, Mirzoeff highlights the notion that cross-disciplinary endeavors involve the identification of a central theme, subsequently orchestrating the convergence of two or three distinct disciplines around it. This interdisciplinary synergy aims to generate an innovative intellectual construct, one that transcends the boundaries of individual disciplines (*Mirzoeff, 1999*). As a result, the concept of visual communication has progressively evolved into an indispensable framework within the domains of art and design. *Vigoroso, Caffaro & Cavallo (2020)* introduces visual communication and design principles, including perception, composition, typography, colour, and rhetoric. It also explores how graphical images are used in various contexts, such as advertising, journalism, art, and film. *Bringhurst (2004)* discusses typography's historical and cultural significance and includes examples of effective design from different historical periods and cultures. *Henderson (1991)* present a framework for understanding how graphic desigfn affects reading based on empirical. It explores the effects of typography, layout, colour, and imagery on reading speed, comprehension, and memory. *Houts et al. (2006)* reviews the research on the design of visualizations for effective communication, covering topics such as perception comprehension. The study discusses the factors that affect the effectiveness of visualizations, such as visual encoding, structure, and interaction, and offers recommendations for designing compelling visualizations (*Houts et al., 2006*). *Richardson, Drexler & Delparte (2014)* reviews the literature on graphic design principles for e-learning, including topics such as typography, colour, layout, and multimedia. The study discusses the implications of these principles for the design of effective e-learning materials and makes recommendations for integrating visual design into the e-learning design process (*Richardson, Drexler & Delparte, 2014*).

Lastly, the incorporation of domain-specific knowledge or constraints into deep learning models can be explored. Design principles and expert knowledge play a vital role in poster design, and leveraging such information during the clustering process could lead to more

meaningful and context-aware results. Integrating domain-specific knowledge with deep learning models could enhance the quality and relevance of the clustering outcomes. Through the above study, it is easy to see that as visual communication theory continues to expand, people are applying it to various fields of art design, no longer just limited to the creation of traditional media mediums. We can obtain more information by refining the various elements in typographic design.

## Sentiment element clustering

As artificial intelligence technology advances, the desire for abstract features has grown beyond the traditional visual elements. Researchers must now consider the potential of arranging and combining features of a single part to create new abstract features. Therefore, it is crucial to analyze the emotions expressed in art design using deep learning techniques, and deep learning has a wide range of applications in such research. The deep self-encoder model proposed by *Hinton & Salakhutdinov (2006)* is a feature extraction model with excellent performance; the algorithm maps high-dimensional data to low-dimensional space by an encoder and then recovers the original data by the decoder to reconstruct the low-dimensional space data features in an unsupervised training manner. When the model converges, the encoder can achieve the role of feature extraction and data dimensionality reduction. *Vincent et al. (2010)* proposed a stacked denoising autoencoder model to obtain a more robust algorithm for stability. The model uses data with noise instead of the original data to train the autoencoders, and the resulting model has more substantial generalization and noise immunity than the conventional autoencoders. Since deep models have more vital feature mining ability than shallow models, the stack denoising self-encoder model increases the depth of the model by cascading multiple denoising self-encoders. In addition, the stack-denoising self-encoder uses a layer-by-layer training method to alleviate the problem of a complex convergence of the depth model.

The Stochastic Neighbor Embedding (t-SNE) algorithm was proposed by *Der Maaten & Hinton (2008)* to address the problem that traditional distance metric algorithms have difficulty characterizing the differences between high-dimensional data. The algorithm offers a novel distance metric instead of Euclidean distance. First, a probability vector is constructed for each category the data point belongs to based on the distance from the data point to each centre of mass. Then a probability vector is built similarly in the feature space after data dimensionality reduction. The clustering model is optimized by minimizing the two probability vectors' relative entropy (Kullback–Leibler Divergence, KL). *Xie, Girshick & Farhadi (2016)* proposed a combined deep embedded clustering (DEC) model combining stacked denoising self-encoder and t-SNE algorithm. The model obtains an encoder capable of feature extraction by greedily training a stack denoising self-encoder layer by layer. Then, the encoder parameters are fine-tuned to optimize the loss of t-SNE and complete the clustering. *Guo et al. (2017)* proposed optimized deep embedded clustering based on the model IDEC model, which uses the loss of the self-encoder and the loss of clustering on feature data as multiple loss functions to train the network, thus preventing the fine-tuned self-encoder network from the original self-encoder feature extraction space from being corrupted.

While deep learning methods have shown immense potential in various applications, including poster design clustering, there are still certain aspects that can be improved compared to the method proposed in this article.

One aspect that requires attention is the computational complexity associated with deep learning models. Deep clustering algorithms often involve training large-scale neural networks, which can be computationally expensive and time-consuming. Improving the efficiency and scalability of these models is crucial to enable faster data clustering, particularly in scenarios where large datasets or real-time clustering are involved. Another area for improvement lies in the interpretability of deep learning models. Deep clustering methods often rely on complex architectures and numerous parameters, making it challenging to interpret and understand the underlying reasoning behind the clustering results. Enhancing the interpretability of deep learning models would provide designers and researchers with valuable insights into the clustering process and help them make more informed decisions.

Furthermore, the robustness and generalization capabilities of deep learning models can be improved. It is essential to ensure that the proposed clustering framework performs consistently well on diverse datasets and is not overly sensitive to variations in input data or minor changes in the design elements. Robust models would provide reliable and stable clustering results across different scenarios and design styles.

The above research shows that deep learning methods have made good research progress in clustering. For the more abstract emotional elements of the art design, the use of deep clustering to complete the given poster with fast data clustering is of great significance for future intelligence and rapid innovation.

# SENTIMENT CLUSTERING MODEL BASED ON CNN AND VAE

## Image processing under convolutional neural network

CNN is a deep learning model commonly used for image recognition and computer vision tasks (*Shi et al., 2022*). It progressively extracts feature information from input images through different hierarchies such as convolutional layers, pooling layers and fully connected layers to achieve tasks such as classification, localization and recognition of images (*Bhatt et al., 2021*).

The core of CNN is the convolution operation, the basic idea of which is to slide a convolution kernel (also known as a filter) over a picture, weigh it, and sum it with the corresponding pixel values in the image to obtain a new feature map. The convolution operation helps the network to capture the local features in the picture, thus enabling feature extraction for the whole picture. Specifically, the convolution operation can be expressed as the following equation (*Lin et al., 2022*):

$$y_{i,j} = \sum_{m=1}^{M} \sum_{n=1}^{N} w_{m,n} x_{i+m-1,j+n-1} + b \tag{1}$$

where denotes the pixel values of row $y_{i,j}$ $i$ and column $j$ in the new feature map obtained after convolution; $x_{i+m-1,j+n-1}$ denotes the pixel values of row $i+m-1$ and column $j+n-1$ in

the original image; $w_{m,n}$ denotes the weight values of row $m$ and column $n$ in the convolution kernel; and $b$ represents the bias term.

To reduce the dimensionality and the number of parameters of the feature map, CNNs usually use pooling operations. The pooling operation can count the pixel values of each local region in the feature map to obtain a new pooled feature map. The common pooling operations are maximum and average pooling, as in Eqs. (2) and (3):

$$y_{i,j} = \max\left(x_{i+m-1,j+n-1}\right) \qquad (2)$$

$$y_{i,j} = \frac{1}{MN}\sum_{m=1}^{M}\sum_{n=1}^{N}x_{i+m-1,j+n-1} \qquad (3)$$

where $y_{i,j}$ denotes the pixel values of the row $i$ and column $j$ in the new feature map obtained after pooling; $x_{i+m-1,j+n-1}$ indicates the pixel values of the row $i+m-1$ and column $j+n-1$ in the original feature map; $M$ and $N$ denote the size of the pooled region. By continuously using convolution and pooling operations, the CNN can gradually extract the high-level feature information in the input image, thus achieving tasks such as classification and recognition.

To ensure the robustness of the convolution operation and the ability to accurately detect targets within images, even if they undergo translation, downsampling using pooling layers becomes essential. The utilization of pooling layers serves the purpose of reducing the size of the feature map.

The two commonly employed pooling operations are maximum pooling and average pooling. Maximum pooling selects the maximum value within each rectangular region of the feature map as the output, while average pooling calculates the average value within the region. By applying pooling operations, the spatial information is condensed while retaining the most salient features.

The convolutional and pooling layers are arranged in an alternating fashion and can be stacked multiple times, giving rise to deep networks. The depth of CNN allows for the learning of increasingly complex features, enhancing their ability to recognize and classify images effectively. As the network deepens, it becomes capable of capturing hierarchical representations, enabling it to discern intricate patterns and subtle variations in the input data. At the topmost layer of the network, a fully connected layer typically consolidates the learned elements from previous layers. This layer integrates the extracted features and produces the final classification output. By combining the representations learned throughout the network, the fully connected layer captures the high-level information necessary for making accurate predictions or classifications. The hierarchical arrangement of convolutional and pooling layers, coupled with the final fully connected layer, forms a comprehensive deep learning framework. This architecture empowers CNN to learn intricate and abstract features, leading to improved image recognition and classification performance. The ability to capture complex visual patterns and generalize from learned features contributes to the success of deep CNN in various computer vision tasks.

In this article, we use CNN to extract sentiment element features and complete the clustering analysis of sentiment elements based on the generative model.

## CNN-based clustering of VAE sentiment elements

While traditional clustering methods often encounter many difficulties when dealing with large-scale high-dimensional data, deep learning-based clustering methods are becoming increasingly popular due to their high-dimensional nature. One of the deep learning-based clustering methods is variational autoencoder (VAE), a generative model that enables efficient data compression and image reconstruction through probabilistic coding and decoding techniques (*Hu et al., 2023*). VAE is a self-coding neural network that maps input data into a distribution of potential spaces by an encoder, which a decoder then samples from that distribution to generate data with similar features. In this process, VAE learns a low-dimensional representation of the data by minimizing the reconstruction error and the regularization term in the potential space, which makes the clustering analysis more efficient. The process of cluster analysis using VAE is roughly divided into two steps: first, a low-dimensional representation is obtained by using VAE for dimensionality reduction. Then, the clustering algorithm is used to cluster the reduced-dimensional data to get the clustering results of the emotional elements in art design. Due to the efficiency and excellent performance of VAE, it has been widely used in image clustering, audio clustering, text clustering, *etc.* (*Anvekar et al., 2022*).

VAE aims to minimize the empirical evidence lower bound (ELBO) which consists of two components: reconstruction loss and a potential spatial regularization term. The reconstruction loss represents the reconstruction error on the original data, while the potential spatial regularization term can constrain the distribution of potential variables by increasing the KL scatter. Specifically, VAE maps the input data $x$ into a mean vector $\mu$ and a standard deviation vector $\sigma$ and generates the latent variables by randomly sampling $z$ from a Gaussian distribution $N(\mu, \sigma)$. Thus, the training objective of VAE can be expressed as the following equation:

$$\mathcal{L} = \mathbb{E}_{Z \sim q(z|x)}[\log p(x|z)] - D_{KL}(q(z|x) \parallel p(z)) \tag{4}$$

where $p(x|z)$ denotes the decoder's generative distribution, $q(z|x)$ denotes the encoder's latent variable distribution, $p(z)$ denotes the prior distribution, and $D_{KL}$ denotes the KL scatter, which is calculated as shown in Eq. (5):

$$D_{KL} = -\frac{1}{2}\sum i = 1^n \left(1 + \log(\sigma_i 2) - \mu_i 2 - \sigma_i 2 - ez_i\right) \tag{5}$$

where $e^{z_i}$ is an empirical parameter that is used to limit the range of potential space vectors and is usually set to a small value. During the training process, VAE uses a backpropagation algorithm to calculate the gradient of the loss function and updates the parameters in the network based on the slope. Specifically, VAE uses the stochastic gradient descent (SGD) algorithm or its variants for optimization.

The calculation process of backpropagation mainly involves the solution of the error in each stage, where the output layer error is shown in Eq. (6):

$$\delta_j^L = \frac{\partial C}{\partial a_j^L}\sigma'\left(z_j^L\right) \tag{6}$$

VAE for the emotion element clustering

Input the picture with emotion element X.

2. Encoding: Map x to the mean vector μ and variance vector $\sigma$ in the latent space z.

3. Sampling: Randomly sample a set of potential vectors z from the potential space z and use these vectors to generate a reconstructed image x '.

4. Decoding: Generate a new image x by taking the sampled latent vector z and the reconstructed x '.

5. loss function calculating and model training: training the model using the backpropagation shown in Eq(6)-(9)

6. Generation: generating pseudo data samples using generated data

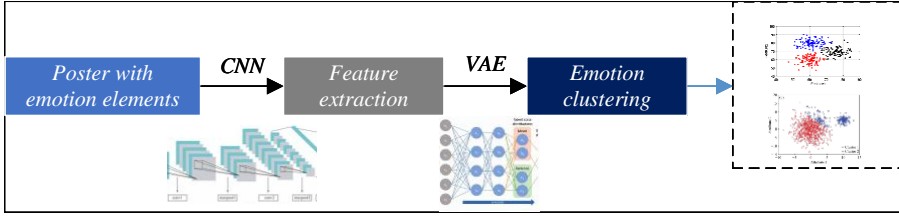

**Figure 1** **The framework for the emotion clustering for the poster design.** The proposed poster design framework with the ability to cluster emotional elements is shown.

where $C$ is the loss function, $a_j^L$ is the activation value of the $j$ neuron in the output layer, $z_j^L$ is the weighted input of the $j$ neuron in the output layer, and $\sigma'$ is the derivative of the activation function. The error of the hidden layer is then.

$$\delta_j l = \sigma'\left(z_j^l\right) \sum_k \delta_k^{l+1} w_{jk} \tag{7}$$

where $l$ it indicates the number of the current layer, $w_{jk}$ is the weight between the neuron in the $kl+1$ layer and the neuron of $j$ in the $l$ layer, and $\delta_k^{l+1}$ is the error of the neuron $k$ in the $l+1$ layer. Based on this, we can calculate the bias gradient and weight gradient to complete the backpropagation calculation of the model error; for reducing the model error, the bias gradient and weight gradient are calculated as shown in Eqs. (8) and (9):

$$\frac{\partial C}{\partial b_j l} = \delta_j l \tag{8}$$

$$\frac{\partial C}{\partial w_{jk} l} = a_j l \delta_k^{l+1} \tag{9}$$

Therefore, the classification process of emotional elements in art design using the VAE method in this article is shown in the algorithm:

The poster design framework with the ability to cluster emotional elements proposed in this article is shown in Fig. 1.

This article delves into the exploration of abstract emotional elements within the realm of visual communication art design. To begin with, a key focus is placed on leveraging

the capabilities of CNN to accomplish the crucial task of feature extraction from the informational content of poster images. By employing CNN, the emotional elements embedded within the visual designs are effectively captured and represented. Subsequently, these extracted features serve as the foundation for conducting a comprehensive cluster analysis of the emotional elements, employing the powerful VAE methodology. Through this approach, a deeper understanding of the underlying emotional characteristics and patterns in the poster designs is achieved, paving the way for innovative insights and advancements in the field of visual communication art design.

## EXPERIMENT RESULT AND ANALYSIS

In this article, the corresponding posters were selected for data analysis in the experiment, and the sentiments involved were clustered by the model proposed in Chapter 3. The dataset consists of 10,000 digital images of art design posters. The dataset is comprised of RGB images in JPEG format. The Art Design Poster Dataset is a collection of diverse posters created by artists from various genres and styles. The dataset aims to cover a wide range of artistic designs, including abstract art, typography, illustrations, photography, and more. Each poster in the dataset represents a unique artistic expression.

The posters in the dataset vary in terms of content, color palette, layout, and overall visual composition. They may contain textual elements, images, or a combination of both. The dataset is carefully curated to include a broad spectrum of artistic styles, ensuring that the CNN model trained on this dataset can learn to recognize and analyze different design elements and patterns. The dataset is accompanied by annotations that provide additional information about each poster. These annotations may include metadata such as artist name, year of creation, genre, and any other relevant details that could assist in analysis or categorization tasks.

In the model evaluation, we use the silhouette coefficient, a standard coefficient for evaluating the clustering effect, as shown in Eq. (10).

$$sc(i) = \frac{b(i) - a(i)}{\max a(i), b(i)} \tag{10}$$

where. $a(i)$ represents the data $i$ is the average distance to other data. $b(i)$ represents the average distance from the data $i$ the average distance to other clusters. The larger the contour coefficient is, the more obvious the data clustering effect is. In this test, the selected poster sentiment elements are mainly divided into three categories, which are positive, negative and neutral.

### The effect of sentiment clustering

To generate images, we conducted data clustering analysis based on the data feature points, and different colour data represent the related emotional performance. The clustered results are shown in Fig. 2.

Considering the privacy of the data and the copyright characteristics, we only show the feature clustering results under the two classes of features extracted from the data. It can be seen that after the VAE method, the three classes of components can be distinguished,

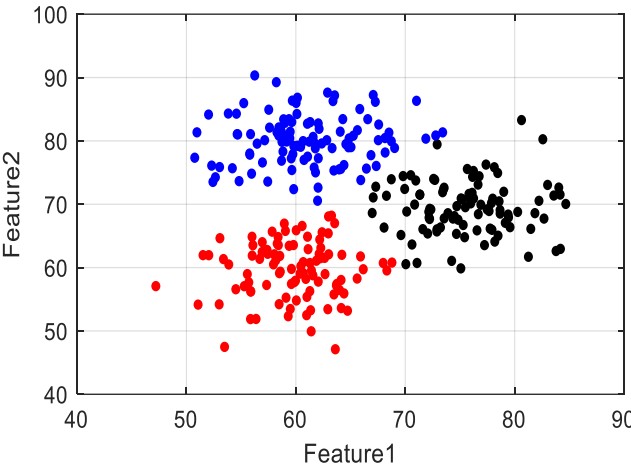

**Figure 2** **The clustering result for the emotion clustering.** To generate images, we conducted data clustering analysis based on the data feature points, and different colour data represent the related emotional performance. The clustered results are shown.

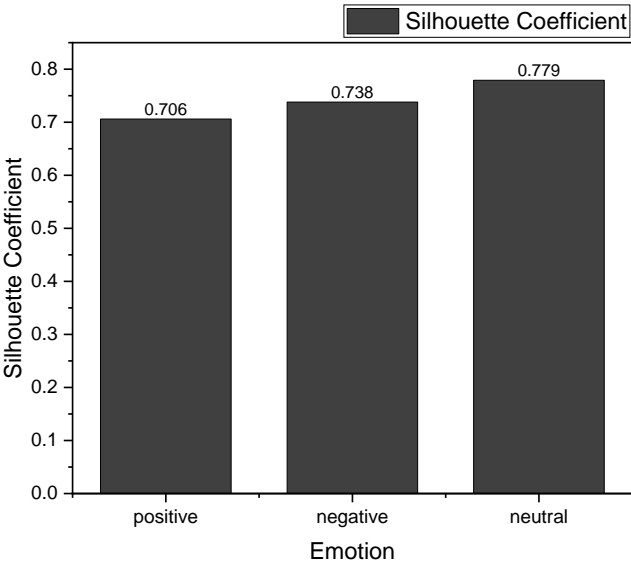

**Figure 3** **The silhouette coefficient for the clustering result.** According to the results, the CNN-VAE framework used in this study has a more balanced performance in the clustering of three types of emotions, and the silhouette coefficient is over 0.7, which indicates that its clustering effect is better. Meanwhile, in order to illustrate the clustering effect more intuitively, we tested its precision in the subsequent experiments.

and to evaluate the effect of clustering, we calculated the silhouette coefficient, the results of which are shown in Fig. 3.

According to the results in Fig. 3, the CNN-VAE framework used in this article has a more balanced performance in the clustering of three types of emotions. The silhouette

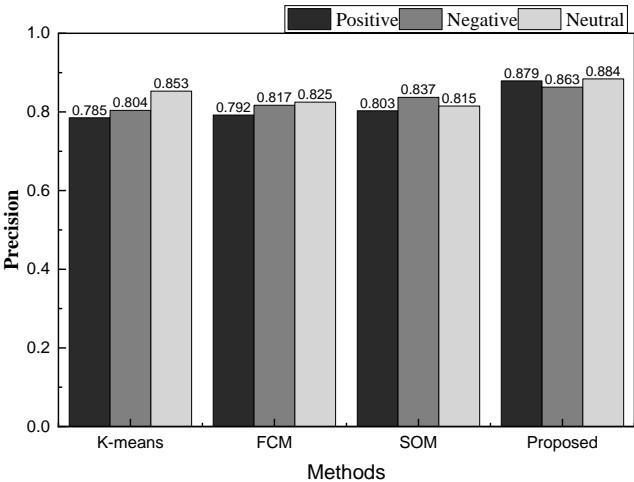

**Figure 4** **The method comparison in clustering precision.** In order to verify the clustering effect, we selected three more classical clustering algorithms in clustering methods for comparison, namely K-means, FCM and SOM, and the real label comparison accuracy under different methods is shown.

coefficient is over 0.7, which indicates that its clustering effect is better. Meanwhile, we tested its precision in the subsequent experiments to illustrate the clustering effect more intuitively.

## Comparison of different clustering methods

To verify the clustering effect, we selected three more classical clustering algorithms in clustering methods for comparison, namely K-means, FCM and SOM, and the accurate label comparison accuracy under different ways is shown in Fig. 4.

From Fig. 4, we can find that the accuracy is significantly higher than that of the traditional method after clustering, and the recognition of the three types of emotions is also more balanced. Hence, the model has great potential for future applications.

## Practical analysis of clustering effects

To better verify the actual effect of clustering, we made artificial poster modifications in the real test by adding the typical emotional elements in 4.1 to the designed poster, whose accuracy after clustering is shown in Fig. 5.

Through the data in Fig. 5, we can find that the proposed framework achieves good accuracy in all three categories of sentiment clustering, in which the success rate of clustering posters involving negative elements reaches 87.9%, and its clustering ability is fast and more accurate for art designs with specific characteristics.

## Framework operation efficiency testing

Because in practical applications, designs containing elements of a particular class are often analyzed quickly, it is crucial to cluster a single kind of sentiment element. In this article, a system operation efficiency test for a single sentiment clustering was conducted, and the results are shown in Fig. 6. Execution Time: This metric measures the time taken by the

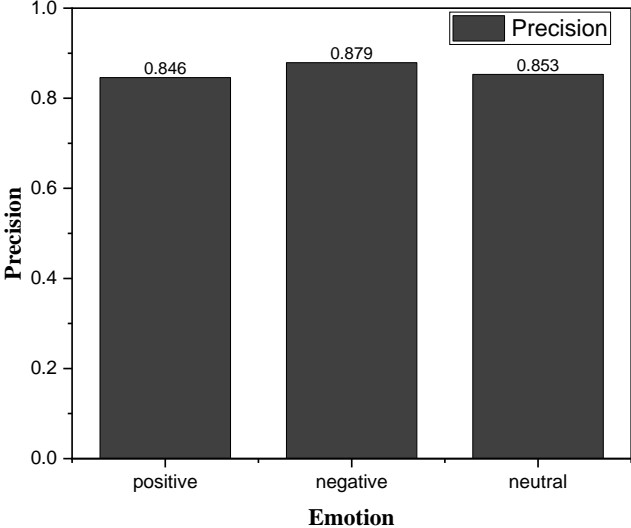

**Figure 5 Precision for the emotion element clustering.** In order to better verify the actual effect of clustering, we made artificial poster modifications in the actual test by adding the typical emotional elements in 4.1 to the designed poster, whose accuracy after clustering is shown.

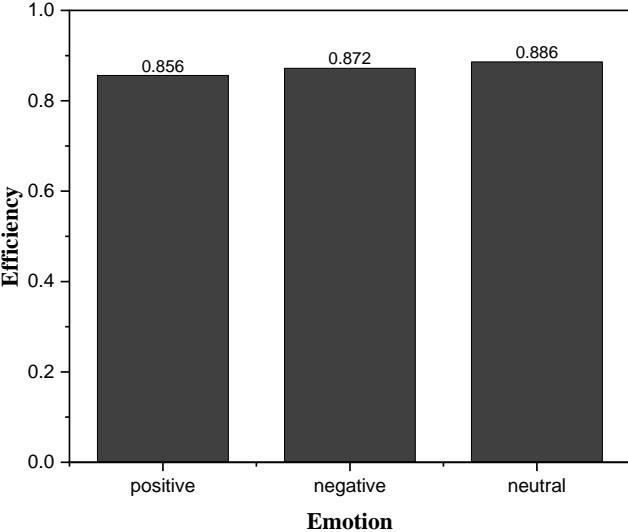

**Figure 6 The operating efficiency for one kind of emotion clustering.** Considering the fact that in practical applications, designs containing elements of a certain class are often analyzed quickly, it is crucial to cluster a single kind of sentiment elements. A system operation efficiency test for a single sentiment clustering was conducted, and the results are shown.

framework to execute a particular task or operation. It can be calculated by measuring the time difference between the start and end of the operation.

During the evaluation process, when focusing solely on a single sentiment element, the model consistently demonstrated impressive efficiency rates surpassing 85% across all

test cases. This noteworthy performance attests to the model's effectiveness and efficiency when applied specifically to a single sentiment element. Such high efficiency holds great significance for the widespread application of the model in future endeavors. By excelling in the analysis of individual sentiment elements, the model's capabilities align with the demand for targeted and specialized sentiment analysis tasks. The ability to efficiently process and analyze sentiments in isolation opens up a multitude of possibilities for diverse applications, spanning various domains such as market research, social media analytics, and sentiment-driven decision-making processes. As a result, the model's exceptional efficiency in handling single sentiment elements represents a valuable contribution towards advancing sentiment analysis techniques and their practical deployment across a wide array of real-world scenarios.

## DISCUSSION

This article delves into enhancing art design elements in the context of optical transmission. Given that conventional visual features like colour and layout no longer fulfil the design requirements of the present era, this article presents a deep learning-based clustering approach for emotional elements in posters. Typically, traditional clustering algorithms traverse all data in every iteration, making them computationally expensive and inefficient when dealing with large data volumes, especially in image data containing various information elements that can't be directly applied to clustering methods for rapid recognition. Therefore, this article adopts a CNN for feature extraction, followed by clustering. The deep clustering algorithm decreases data dimensionality by extracting data features while retaining them. It provides a more efficient and essential data characterization that enhances the ability to fit data correlations and reduces iterative computation. However, commonly used autoencoders have the drawbacks of weak anti-interference ability and poor generalization ability, while generative adversarial networks have the disadvantages of insufficient models and poor convergence (*Abir et al., 2023*). Hence, this article proposes a CNN-VAE-based clustering method that is expected to rapidly and precisely cluster emotional elements in visual communication, aiding designers in classification during the design process.

Art and design are not independent phenomena but are influenced by various factors such as post-industrial technology culture, post-modern visual culture, regional and ethnic cultures, and other related social cultures. It presents new cultural developments in design, and cultural pluralism promotes the diversified development of design forms and concepts. Any "monocultural tendency" is unsustainable, as it goes against the law of development of design cultural pluralism and creates many problems (*Hou et al., 2022*). The influence of modern technology, such as the Internet, digital virtual, and artificial intelligence, has made visual information dissemination and access easier and more convenient. Still, it has also weakened the cultural and human subjectivity of design. Therefore, it is worth exploring whether design innovation that solely relies on technological development meets the current needs of visual humanities and the direction of future design development.

With the rapid improvement of technology and productivity, especially with the popularity of computer and network technology, information technology has gradually

replaced traditional industrial technology production methods as the new driving force of social development. Social life and behaviour patterns have undergone significant changes, and social culture has been transformed with the arrival of the information age. In the field of visual communication design, the intervention of technological elements has become an inevitable requirement to meet the needs of society. The perfect combination of modern technology and visual arts is crucial (*Li, Wang & Xu, 2022*). In the information age, the globalization of information has made information exchange easier, and the integration of cultures has led to the ''generalization'' of visual communication design forms. Therefore, in the future of visual communication art design, it is necessary to leverage the advantages of artificial intelligence technology to accelerate the design process and improve quality.

## CONCLUSION

The authors of the article have examined the emotional elements in the art design process within the context of visual communication. They have introduced a novel framework that involves pre-processing image data, extracting emotion features using CNN, and clustering the data using VAE. The experimental results demonstrate strong performance, with a clustering silhouette coefficient exceeding 0.7 for all three sentiments and a clustering precision exceeding 80% in actual testing. The proposed framework's contribution lies in its ability to effectively analyze and cluster graphic design products based on emotional symbols. By combining feature extraction techniques and clustering algorithms, the framework provides valuable insights into the emotional aspects of visual communication in art design. Furthermore, the article highlights the operational efficiency of the framework, with the clustering of single emotion elements achieving an efficiency of over 80%. This efficiency metric showcases the framework's effectiveness in handling image clustering tasks, which is a crucial aspect for practical application. In conclusion, the article provides a theoretical and practical foundation for future research in examining abstract factors within visual communication in art design using artificial intelligence methods. The authors could further emphasize the novelty and potential impact of their proposed method on the field, underscoring its significance in advancing the understanding and application of emotional elements in art design.

However, the CNN and VAE methods used in this article rely on network structure and parameter adjustments based on prior experience to enhance the convergence and robustness of the model, which require further enhancement in future research. In addition, for clustering high-definition images, reducing the complexity of operations and improving operational efficiency will require further investigation.

### Funding

The author received no funding for this work.

### Competing Interests

The author declares that there are no competing interests.

### Author Contributions

- Xue Rui conceived and designed the experiments, performed the experiments, analyzed the data, performed the computation work, prepared figures and/or tables, authored or reviewed drafts of the article, and approved the final draft.

### Data Deposition

The code is available in the Supplementary File.

The data is available at Kaggle and Zenodo:

https://www.kaggle.com/datasets/phiitm/movie-posters

https://doi.org/10.5281/zenodo.8108642.

### Supplemental Information

Supplemental information for this article can be found online at http://dx.doi.org/10.7717/peerj-cs.1548#supplemental-information.

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
