# Peer review of "A convolutional neural networks based approach for clustering of emotional elements in art design"

_PeerJ Computer Science, doi:10.7717/peerj-cs.1548_

## Round 0.1 · original submission · Minor Revisions

Dear authors,

Thanks for your submission to our esteemed journal, your paper has been reviewed by the experts in the field and you will see that they have a couple of improvements to be done. I agree with them and also of the view that you must also improve the structure and grammar of the manuscript during revision. Therefore we invite you to revise and resubmit your article after carefully updating.
Thanks.

Reviewer 1 ·

Basic reporting

A. In this paper, the emotional elements in the process of art design under the background of visual transmission are analyzed and studied. Through the feature extraction of graphic design product pictures with emotional signs, the cluster analysis of its expression of emotion is completed. The proposed framework extracts the emotional features in the image data through CNN after image preprocessing and then uses these features to complete the clustering analysis through VAE.

B. The language expression requires some modifications. Please see the detailed comments;
C. Basically, you should enhance your contributions, and limitations, underscore the scientific value added of your paper, and/or the applicability of your findings/results and future study in the Introduction;
D. In Section 2, I learned that "using deep clustering to complete fast data clustering of the given posters is of great significance for future intelligent design and rapid design," but what aspects of current deep learning methods need to be improved compared with the method proposed in this paper?
E. The title of each part in the article structure needs to be further summarised and refined;
F. Add a new paragraph to introduce Figure 1;
G. In this paper, the corresponding posters are selected for data analysis in the experiment, but the description of the dataset (size, type, etc.) is missing;
H. The simultaneous appearance of red and green colors in Figure 2 should be avoided;
I. In Section 4.4, Framework operation efficiency testing, How are the metrics calculated?
J. The conclusion provides a good summary of the paper's contributions and future research directions. However, the authors could benefit from more explicitly highlighting their proposed method's novelty and potential impact.

Experimental design

See above

Validity of the findings

See above

Additional comments

See above

Reviewer 2 ·

Basic reporting

This study delves into exploring abstract affective elements within the visual communication art design methodology domain. Firstly, the authors have implemented a Convolutional Neural Network (CNN) that is utilized to extract effective features from the information depicted in poster images. Subsequently, these extracted features are employed in conjunction with a Variational Autoencoder (VAE) to conduct a clustering analysis on the affective elements. This clustering analysis categorizes the poster images into positive, negative, and neutral clusters based on their affective content. By leveraging artificial intelligence technology, this research facilitates the extraction of multidimensional abstract data, thus expediting the art design process.
The research topic selected by the author(s) is important, the overall structure and organization of the paper are satisfactory, and the paper qualifies for an average up-to-date bibliography. However, there are some suggestions that should be considered to improve the quality of the paper.

(1) Within the introduction, the author initiates a discussion on the impact of color clustering on visual design, which serves as an effective starting point. However, the subsequent analysis pertaining to this aspect is conspicuously absent in the results section.
(2) It would be highly beneficial to witness the practical application and ensuing discussion of this research.
(3) “Paul introduces visual communication and design principles……”,The added references don't seem to correspond.
(4) In the literature review Section, the author should add more comments on the listed literatures (Section 2.1).
(5) Additionally, a comprehensive exposition elucidating the author's planned implementation of the method should be provided.
(6) Please furnish additional intricate details regarding the theoretical underpinnings to bolster and enhance the elucidation of these formulas.
(7) Despite the extensive corpus of state-of-the-art research encompassing disparate approaches posited by various scholars, it regrettably falls short of distinctly and unequivocally delineating the research gap.
(8) The explication of this gap, which the author aims to address in a more superior manner compared to the solutions proffered by other researchers, appears somewhat ambiguous to the reader.
(9) The paucity of references is conspicuous, thus I would recommend that the authors augment the list and incorporate more pertinent and contemporary references within the text.It would be advantageous to supplement the work with references from esteemed journal publications.

Experimental design

No Comments

Validity of the findings

No Comments

---

## Round 0.2 · accepted · Accept

We are pleased to inform you that your paper has been recommended for publication based on the experts' comments and my review. Thanks for your fine contribution and good luck with your future research.

Reviewer 1 ·

Basic reporting

The authors have addressed all the comments raised by me.

Experimental design

The authors have addressed all the comments raised by me.

Validity of the findings

The authors have addressed all the comments raised by me.

Additional comments

The authors have addressed all the comments raised by me.

Reviewer 2 ·

Basic reporting

Authors have addressed all the issues.

Experimental design

Well-desinged.

Validity of the findings

Well stated conclusions.